# Modelling students' cognitive achievement skills using the alpha power transformed Lindley probability distribution

Shibiru Jabessa Dugasa[1]*, Butte Gotu Arero[2]

1 Department of Mathematics, Kotebe University of Education, Addis Ababa, Ethiopia, 2 Department of Statistics, Addis Ababa University, Addis Ababa, Ethiopia

* shibirujabessa2021@gmail.com

## Abstract

Cognitive achievements in mathematics skill scores are crucial for daily life in modern society. The objectives of this study were to apply the alpha power transformed Lindley probability distribution to students' cognitive achievement skill scores using regression models and to identify the best probability distributions for cognitive achievement, including APTLD, using Young Lives datasets. This study proposes regression modeling using the alpha power transformation Lindley probability distribution for the application of cognitive achievement in mathematics skill scores. The study found that students' average mathematics skill score was 37.01%, with a standard deviation of 14.9, reflecting performance variation. Parental education differed significantly, with 48.5% of mothers and 34% of fathers lacking formal schooling. Additionally, 59% of students lived in rural areas, while 41% resided in urban settings. The average household size was 5.77 members, showing variability in family structures. From the results, the findings show that the mean cognitive achievement in mathematics skill scores (37.01) is greater than the median (33.33), indicating that the data are positively skewed or right-skewed. The APTLD regression model demonstrates the best fit for the data, as indicated by its lowest AIC and BIC values compared to the APTEPLD, TPLD, and TwPLD models. This confirms its superiority in capturing the underlying structure of mathematics skill scores, making it the most suitable model for analyzing cognitive achievement. Therefore, this new model can be considered a significant contribution to the field of statistics and probability methods. Future work on the presented study could extend the APTLD distribution using Bayesian regression models.

## Introduction

Education is defined as the intentional endeavor to pass on acquired experience, knowledge, and abilities from one generation to the next. This can occur formally

**Data availability statement:** The dataset used in this study is freely available and can be downloaded by providing a justification the reason for requesting the data from the link: https://ukdataservice.ac.uk/deposit-data.

**Funding:** The author(s) received no specific funding for this work.

**Competing interests:** The authors have declared that no competing interests exists.

within the walls of a school or informally outside the classroom. The goal is to prepare learners to cope with and meet the challenges of their society. The methodical process of education aims to provide people with the cognitive and motor abilities necessary to tackle life's challenges. In essence, education aims to prepare individuals to be useful members of their society. When properly managed, education has the power not only to develop individuals but also to advance a nation's social, economic, political, cultural, and technological aspects [1].

Cognitive skills, sometimes called intelligence, include students' situational and intellectual quotients. Students' cognitive abilities have been assessed over time based on their academic performance on internal and external exams [1]. Cognition primarily involves memory, especially in early childhood, and the development of personal thoughts and beliefs about the world. Students' cognitive skills play a crucial role in their academic success and overall educational attainment, as these skills facilitate the learning process [2]. Research indicates that intelligent students possess present, comprehensive understanding of the subject, demonstrate a desire to study, and demonstrate a passion for the subject. Cognitive skills, also known as cognitive functions, abilities, or capacities, are the mental processes involved in gathering and processing information to perform tasks [1,2].

According to [2], academic achievement is defined as the real performance of students in mastering academic knowledge and ability, as illustrated based on examinations following systematic learning. In the Chinese educational evaluation system, universities typically branched students based on their academic achievement. The authors [1,2] argue that academic achievement, particularly in China's college entrance exams, determines students' future development. Therefore, studying the factors that enhance academic achievement is crucial for each student's learning and development. For instance, in China's educational selection system, academic achievement is the primary measure and the most important reference factor for college and university admissions [2]. A cognitive achievement in mathematics is crucial for daily life in modern society. It influences academic success from primary school, particularly in STEM (science, technology, engineering, and mathematics) subjects, through to higher education [3]. This foundation leads to better employment opportunities, improved quality of life, informed decision-making [4], and better health outcomes [5]. Researchers have emphasized the developmental changes in mathematics, noting that acquiring mathematical skills involves gradually absorbing knowledge throughout an individual's schooling [6]. For instance, the ability to solve math problems automatically is essential for tackling more complex calculations and forms the basis for advanced mathematical understanding [7–9].

Numerous studies have investigated mathematics performance in primary school [10–13], indicating that both domain-specific (knowledge that are tailored to a particular area or field) and domain-general (abilities that are applicable across various domains) factors predict mathematics achievement. Cognitive proficiency in mathematics is essential for daily life and greatly influences academic success, particularly in STEM fields. This proficiency leads to better job prospects, improved decision-making, and enhanced health outcomes. Mathematical skills can be

developed progressively throughout schooling, with foundational abilities like automatic problem-solving serving as the groundwork for more complex calculations and advanced mathematical comprehension [3–5].

Children's numerical representations develop through their experiences and formal mathematical education. This development is closely linked to the growth of specific and general cognitive mechanisms, such as attention, cognitive control, working memory, and environmental factors. Studies indicate that both domain-specific and domain-general factors predict mathematics achievement in primary school [6–8,10].

According to the Organization for Economic Co-operation and Development (OECD), the Better Life Index (BLI) encompasses 11 topics: income, jobs, housing, community, safety, education, environment, civic engagement, life satisfaction, health, and work-life balance. Each topic is evaluated using one to three indicators. For instance, the education topic is measured by indicators such as educational attainment, student skills, and years of education. The OECD defines educational attainment as the highest level of education completed within a country's education system. This metric is frequently studied and often correlated with other indicators in the literature [9].

Researchers have extensively explored the relationship between educational attainment and various health outcomes. For instance, authors [10] identified a link between education and health in older adults, even after considering individual and family characteristics, suggesting that education independently affects health. The study in [11] examined the correlation between educational attainment and suicide rates in the U.S. in 2001. Authors [12] investigated whether self-rated health improved over time for both genders from 1972 to 2002, using educational attainment to explain the trends. Additionally, [13] explored how educational attainment influences perceptions of safety, job satisfaction, adherence to safety policies, and the frequency of accidents.

The author study on [14] examined the influence of human capital on health across 23 countries using data from the OECD Survey of Adult Skills, emphasizing the cognitive aspect of social capital. [15] Analyzed trends in the educational attainment of young male homicide victims in Chicago from 2006 to 2015. [16] Applied regression analysis to explain long-term trends in self-rated health in the U.S., taking into account factors such as gender, age, race, and education.

Modeling real-world phenomena using probability models is essential for making statistical inferences. This field has been extensively researched by statisticians and continues to be an active area of study. The APTL distribution, known for its flexible probability density function (PDF), has led to the development of new alternative models on the unit interval, potentially yielding better results in statistical inference. Many of these distributions are derived by transforming a parent distribution and often outperform the beta distribution in modeling. Additionally, alternative regression models have been created for some of these distributions. The beta regression model was proposed by [17], the Kumaraswamy quantile regression model by [18], and the unit-Lindley regression model by [19]. Reference [20] introduced a regression model based on the weighted-exponential distribution as an alternative to the beta regression model. A flexible regression model for the beta and simplex regression models was developed by [21]. The unit-Weibull quantile regression model, introduced by [22], serves as an alternative to the beta and Kumaraswamy regression models. Statistical distributions serve as essential tools for representing various data characteristics, including right or left skewness, bi-modality, and multi-modality. These features are commonly observed across numerous applied sciences, such as engineering, medicine, finance, and other disciplines [23,24].

In numerous real-world scenarios, classical distributions often fail to accurately represent observed data. As a result, there has been growing interest in creating more adaptable distributions by enhancing classical models with additional shape parameters. Over the past two decades, various generalized distribution families have been introduced and extensively analyzed for data modeling across multiple fields, including economics, engineering, biology, environmental science, medical research, education, and finance [25]. Recent research indicates that the coefficient of kurtosis can take both negative and positive values, implying that the distribution has a flexible structure in data modeling. Additionally, the new distribution appears to be flatter than the normal distribution. As $\theta$ increases, the kurtosis coefficient rises while the variance decreases [23].

The purpose of generating APTLD is to provide new opportunities for modeling various data characteristics, such as left or right skewness, excess kurtosis, and bathtub failure rates. Previous well-known distributions are insufficient for accurately modeling these types of datasets. Specific variants of APTLD can be applied to skewed and long-tailed datasets, improving modeling accuracy by incorporating an additional shape parameter. Furthermore, the newly derived APTLD [26] is highly effective in modeling learning skill scores (mathematics skill scores) with certain covariates within the APTLD regression framework.

This study aims to apply APTLD to students' cognitive achievement skill scores using regression models, identify the best probability distributions for cognitive achievement in mathematics— including APTLD— and analyze the Young Lives datasets. It introduces a unique unit competitive distribution and its regression modeling, focusing on mathematics skill scores and their relationship with various covariates/ explanatory variables. By applying a novel transformation of APTLD, the study compares it with other probability distributions, highlighting the need for new models as current ones may not fit all data types.

## Data and statistical models

This section presents the study design, data collection procedure, and statistical models.

## Data and study design

About 12,000 children in Ethiopia, India, Peru, and Vietnam are the subjects of the observational cohort research Young Lives (YL). In each nation, two cohorts of children, one younger and one older, were recruited and monitored [3]. And this study examined only children from Ethiopia, who were enrolled in 2016. The students were selected from 20 sentinel sites of Ethiopia's five major regions (Tigray, Amhara, Oromia, SNNP, and Addis Ababa) that represent the total population [26,27].Students with missing data on outcomes and co-varieties at the individual level were dropped. Outliers in outcomes based on the WHO standards [28] were also dropped.

The dataset used in this study is freely available and can be downloaded by providing a justification the reason for requesting the data from the link: https://ukdataservice.ac.uk/deposit-data/. Details on the sample, variable construction, and other related information are also provided [29]. Thus, a total sample of 1726 children was used in this study.

## Statistical models

**Alpha power transformed Lindley distribution (APTLD).** The alpha power Lindley (APL) four-parameter probability distribution is an extended family of probability distributions that offers greater flexibility in modeling complex datasets. The distribution is characterized by four parameters ($\alpha$, $\theta$, $\beta$, and $\delta$) that allow it to capture different shapes and behaviors of data. Its flexibility makes it applicable in various fields, including education, health, engineering, survival analysis, and reliability [30].

APTLD is applicable in the area of educational research to model data with positive skewness and lower bounds. It is particularly useful for modeling data such as test scores, study hours, or educational achievement, where students' scores or progress follow a skewed distribution. By using the parameters of the distribution, researchers can better understand the distribution of student performance and design interventions that account for the heterogeneity in student outcomes [30].

APTLD can be used to model student test scores, which are often positively skewed, meaning most students score near the lower end of the scale, with fewer students achieving higher scores. Its shape and location parameters can model this type of data more effectively than other probability distributions like standard normal, exponential, gamma, beta, log-normal, Weibull distributions, etc. The parameters $\alpha$ (shape) control the skewness of the data, $\theta$ (rate) controls the spread or scale of the test score, and $\delta$ (location) shifts the distribution along the x-axis and is used when the test scores are bounded below (minimum score) [31].

APTLD can be applied to modeling study-hours, which often follow a skewed distribution, where most students may study for only a few hours, with some students spending much more time. It can be useful to model the distribution of study hours, taking into account the fact that the data might be concentrated around low values but can exhibit significant variability. The parameter α reflects how concentrated the study hours are at lower values, θ determines the scale of study hours (how long students generally study), and δ can be used to shift the study hours to ensure no negative values [31,32].

APTLD can also be used to modeling educational achievement or growth: In education research, it is often important to model the educational achievement or growth of students over time, such as improvements in test scores or grades. The Alpha Power Transformed Lindley distribution can be applied to model the distribution of the growth of educational outcomes, capturing the heterogeneity of student progress. The parameters α represents how the growth is distributed among students, allowing for the modeling of more concentrated or dispersed achievement changes, θ controls the overall rate of growth, δ shifts the distribution to ensure that the baseline starting score or achievement is realistic [33].

Modeling Dropout Times in Education: The time until a student drops out of an educational program is another potential application of the Alpha Power Transformed Lindley distribution. Educational programs often see a right-skewed distribution of dropout times: most students leave early, but some persist longer. The parameter α captures the shape of the dropout distribution (early dropouts vs. late dropouts), θ captures the rate at which students are expected to leave, and δ allows the distribution to be adjusted to real-life scenarios, where the dropout time cannot be negative. The above authors developed from the three parameter Lindley probability distribution and shows the simulation as well as the application of real life data [31–33]. The Alpha Power Transformed Lindley distribution can be used to model the response variable in the context of mathematics cognitive achievement of students. And, this distribution combines the Lindley distribution (which is a skewed distribution) and an additional alpha parameter to allow for flexibility in the modeling [33].

Let $Y$ be a random variable having APT Lindley probability distribution. The probability density function with transformation parameter $\alpha > 1$, $\beta > 0$, $\theta > 0$, $\delta > 0$ is expressed by the authors can be given as [30]:

$$f_{APTLD}(y_i; \ \beta, \ \theta, \delta, \alpha) = \frac{\log(\alpha)}{(\alpha - 1)} \frac{\theta^2}{(\theta\delta + \beta)} (\delta + \beta y) e^{-\theta y} \alpha^{1 - \left(1 + \frac{\theta\beta y}{\theta\delta + \beta}\right)e^{-\theta y}}$$

(1)

If $Y$ is a random variable having density function (1), we can write $Y \sim APTLD(\alpha, \ \beta, \ \theta, \ \delta)$. The four parameters of the APLD stated above can control the shape and flexibility of the distribution: $\beta$ is scale parameter, $\theta$ is a rate parameter (similar to exponential distributions) $\delta$ is a parameter that shifts or skew the distribution $\alpha$ is a parameter related to the alpha power transformation or shape parameter, and $Y$ is the mathematics cognitive achievement score.

**Exponential family of distributions.** A family of probability mass functions or probability density functions for a random variable $y$ is called an exponential family of distributions if it can be expressed as:

$$f(y; \theta) = c(\theta)h(y) \exp\left\{\sum_{i=1}^{k} w_i(\theta)t_i(y)\right\}$$

Where $h(y) > 0$, and $t_i(y)$ are real valued function of the observation $y$ that do not depend on the parameter $\theta$, $c(\theta) > 0$ and $w_i(\theta)$ are real-valued functions of the parameter $\theta$ that do not depend on observation $y$.

The alpha power transformed Lindley probability distribution (APTLD) is defined in equation (1). The canonical form of an exponential family distribution is given by:

$$f(y) = c(\beta, \ \theta, \delta, \alpha)(\delta + \beta y) exp\{-\theta y\} \alpha^{1 - \left(1 + \frac{\theta\beta y}{\theta\delta + \beta}\right)e^{-\theta y}}$$

where

$$\alpha^{1-\left(1+\frac{\theta\beta y}{\theta\delta+\beta}\right)e^{-\theta y}} = exp\left\{log(\alpha)\left(1-\left(1+\frac{\theta\beta y}{\theta\delta+\beta}\right)exp(-\theta y)\right)\right\}$$

$$f(y) = c(\beta,\ \theta,\delta,\alpha)(\delta+\beta y)\ exp\left\{log(\alpha)\left(1-\left(1+\frac{\theta\beta y}{\theta\delta+\beta}\right)exp(-\theta y)\right)-\theta y\right\}$$

Therefore, the APTLD follows the exponential family distribution

**Generalized linear model.** To derive the linked function in a Generalized Linear Model (GLM) for the context of modeling mathematics cognitive achievement using the alpha power Lindley distribution with regression model, we need to break down the model and its components.

For a GLM, we link the linear predictor $\eta$ to the expected value $\mu$, where, $\mu = E(y)$. Since the exact expected value of the alpha power Lindley distribution is not trivial to compute analytically, we approximate it using a link function. The choice of link function depends on the nature of the distribution and the expected relationship between the linear predictor $\eta$ and the expected value $\mu$. For many distributions, the logit link is commonly used in GLMs.

Regression models are widely used to analyze dependent variables with various covariates [24]. The lifetimes of individuals are generally affected by explanatory variables such as gender, age, alcohol abuse, and smoking. To model these kind of data sets, we propose a new log-location-scale regression model based on the APTLD density. Let $y_i$ be the response variable and $\boldsymbol{x_i}^T = (\ x_{1i}, x_{2i},\ x_{3i}, \ldots, x_{pi})$ is the explanatory variable vector.

Now, let we consider n observations $y_1, y_2, y_3, \ldots, y_n$ from the re-parameterized APTL distribution such that $Y_i \sim APTLD(\eta_i, \alpha, \beta, \delta, \theta)$ with unknown parameters $\eta_i$ and $\alpha, \beta, \delta, \theta$.

$$\eta_i = X_i\beta \tag{2}$$

Where $\beta = (\beta_0,\ \beta_1, \beta_2, \ldots, \beta_p)^T$ and $X_i = (1,\ x_{1i}, x_{2i},\ x_{3i}, \ldots, x_{pi})$ are the unknown regression parameter vector and vector known to covariates. Thus defined, $g(x)$ is the link function which is used to relate the covariates at the PDF of APTLD of the response variable and we adopt the logit-link function such that

$$g(\eta_i) = log\left(\frac{\eta_i}{1-\eta_i}\right) \tag{3}$$

And from equation (2) we can consider the link function below:

$$g(\eta_i) = log\left(\frac{\eta_i}{1-\eta_i}\right) = X_i\beta \tag{4}$$

From Equation (4), it comes

$$\eta_i = \frac{exp(X_i\beta)}{1+exp(X_i\beta)} \tag{5}$$

Now, we need to specify the relationship between the linear predictor $\eta$ and the distribution parameters of the alpha power Lindley distribution. Therefore, a typical of GLM, the mean $\mu$ is a function of the linear predictor, i.e., $\mu = g^{-1}(\eta_i)$ and for the alpha power Lindley distribution, we can use a log link function to model the expected cognitive achievement in mathematics score. Therefore, the link function is the log of the expected value. $\mu = E(y)$ the expected cognitive achievement score $X\beta$ are the fixed effects.

The response variable *y* follows the APT Lindley distribution, adjusted by the total predictor. As a results the final GLM model can be expressed in terms of the random component, systematic component and link function as follows:

$$y_i \sim APTLD(\alpha, \beta, \delta, \theta, \ \eta_i) \tag{6}$$

Where $y_i$ is the expected cognitive achievement in mathematics for student *i* in group *j*.

## Model comparison

To illustrate the practical performance of the APTLD with alternative distributions, we examine data sets in this section. We obtained the data sets from the Young Live data (www.younglives.org.uk). By using maximum likelihood estimators (MLE), we can fit the distribution and compare APTLD with other stated distributions. The model selection is carried out using the Akaike information criterion (AIC) and, Bayesian information criterion (BIC) given by [34]:

$$AIC = -2logL + 2p \tag{7}$$

$$BIC = -logL + plog\ n$$

Where *p* is the number of the model parameters and *n* is the sample size. The model with minimum AIC or BIC value is chosen as the best model to fit the data. We fit the alpha power transformed Lindley distribution (APTLD) to the data sets using MLE and compared with alpha power transformed extended power Lindley distribution (APTEPLD), three parameter Lindley distribution (TPLD) and two parameter Lindley distribution (TwPLD) distributions. APTLD and other models can be implemented by R software using **optim** functions used to estimate the unknown model parameters for regression models, ensuring the alpha power Lindley distribution is used for the response variable (cognitive achievement in mathematics skill score).

## Ethical review statement

This study is based on the Young Lives datasets from the 5th round of household, child, and school surveys, which are publicly archived and available for download from the UK Data Archive. To access and use the data, registration, an application for a password with the UK Data Service, and signing a confidentiality agreement were required. Specific research guidelines can be found at the following link: https://ukdataservice.ac.uk/events/ethical-and-legal-guidelines-in-data-sharing-2025-05-08/. Ethical approval for the data collection was provided by the Central University Research Ethics Committee of the University of Oxford.

## Results and discussion

Table 1 provides the descriptive statistics for the explanatory and response variables. The average cognitive achievement in mathematics skill score was 37.01%, with a standard deviation of 14.9. This indicates that students' mathematics skill scores varied around this mean by approximately 14.9 points. Regarding the mode of transportation of students, about 95.6% of the students made their way to school on foot, while around 4.4% used public buses or other forms of transportation.

Concerning educational levels of mothers, 48.5% lacked formal education, while about 27.2%, 13.1% and 11.2% completed primary school, secondary school, diploma or higher education respectively. This result underscores that about 75% of mothers either lacked formal education or completed primary school. Regarding fathers, about 34% had no formal education, 21% completed primary school, 20% finished secondary school, and 25% achieved a diploma or higher education. These figures reveal a broader range of educational levels among fathers compared to mothers, with a significant

**Table 1. Summary of descriptive statistics.**

| Variables | | Count | Percent |
|---|---|---|---|
| Mode of transportation | Walk | 1650 | 95.6 |
| | Public bus and others | 76 | 4.4 |
| Father's education level | No education | 585 | 33.9 |
| | Primary school | 368 | 21.3 |
| | Secondary school | 337 | 19.5 |
| | Diploma and above | 436 | 25.3 |
| Mother's education level | No education | 838 | 48.5 |
| | Primary school | 469 | 27.2 |
| | Secondary school | 226 | 13.1 |
| | Diploma and above | 193 | 11.2 |
| Access to electricity | Yes | 1107 | 64.1 |
| | No | 619 | 35.9 |
| Place of residence | Rural | 1025 | 59.4 |
| | Urban | 701 | 40.6 |
| Region | SNNP | 431 | 25.0 |
| | Tigray | 341 | 19.8 |
| | Amhara | 317 | 18.4 |
| | Oromiya | 387 | 22.4 |
| | Addis Ababa | 250 | 14.5 |
| **Mean** | | | **Std.dev** |
| Household size | | 5.77 | 1.92 |

portion achieving higher education. Concerning basic services, around 64% of households have access to electricity, which likely enhances living and studying conditions.

The average age of student who participated in the study was approximately 181 months, which is nearly 15 years old. The study included students from four regions and one city Administration: SNNP, Oromia, Tigray, Amhara, and Addis Ababa. The SNNP region had the largest representation, with 25% of the students. Oromia followed with 22.4%, also representing a significant proportion of the students. Tigray and Amhara had nearly similar proportions, with 19.8% and 18.4% of the students, respectively. Addis Ababa had the smallest representation, with 14.5% of the whole students included in the study. Regarding the place of residence, about 59% of the students were from rural areas, while 41% were from urban areas. The data provided indicates that the average household size is 5.77 members, with a standard deviation of 1.92.

Table 2 shows that the mean (37.01) is greater than the median (33.33), typically indicating that the data are positively skewed or right-skewed. This means the mean is pulled to the right (higher values) by a few very large outliers or extreme values, while the median is closer to the lower values because it is less affected by these outliers. The standard deviation of 14.97 suggests considerable variability in students' performance. A skewness of 0.47 confirms the slight right skew, and a kurtosis of 2.53 indicates a distribution close to normal but with slightly fewer extreme values. Overall, the data suggest that most students score around the average, with some higher outliers influencing the distribution.

## Results of univariate data modelling

This section presents the results of parameter estimates and model comparisons. The parameters of the probability distributions provide valuable insights into the characteristics and behavior of the data. For the alpha power transformation Lindley distribution (APTLD), the shape parameter alpha is transformed to ensure it's greater than 1. It controls the

**Table 2. Summary of the cognitive achievement in mathematics score.**

| Mean | 37.01 |
|---|---|
| Standard deviation | 14.97 |
| Median | 33.33 |
| Skewness | 0.47 |
| kurtosis | 2.53 |

skewness and peak of the distribution. A larger alpha can make the distribution more skewed or peaked. Therefore, the results indicates that the shape parameter is equal to 5.74, suggesting that the distribution is likely to be more skewed with a sharper peak.

The scale parameter Beta ($\beta$) affects the spread of the distribution. A higher Beta ($\beta$) leads to a wider spread, indicating more variability in the data. An estimated ($\hat{\beta} = 7.48$) indicates a wide spread in the data, meaning there is considerable variability in the cognitive achievement in mathematics score. Delta $\delta$) can affect the thickness of the tails, influencing how extreme values are handled. The rate parameter ($\hat{\delta} = 11.14$) implies the shape and tail behavior of the distribution. An estimated($\hat{\delta} = 11.14$) suggests a specific shape that may accommodate more or fewer extreme values, depending on its context and relation to other parameters. The rate parameter theta $\theta$) is related to the exponential component of the distribution. A higher theta $\theta$) value leads to a steeper decline, indicating fewer higher values. And, the value of scale parameter ($\hat{\theta} = 2.27$) implies a steeper decline, indicating that high cognitive achievement in Mathematics skill score are less likely.

In contrast, the alpha power transformed extended parameter Lindley distribution (APTEPLD) has lower values for these parameters, suggesting less skewness, a narrower spread, a slower decline in probability, and a lighter tail. The three-parameter Lindley distribution (TPLD) and the two-parameter Lindley distribution (TwPLD) also exhibit moderate spread and tail behavior, with their respective parameters indicating varying degrees of skewness and decline in probability. Overall, these parameters help in selecting the most suitable model for the data and understanding the underlying distribution characteristics.

From the result of Table 3 below, the alpha power transformation Lindley distribution (APTLD) has the lowest AIC (14182.23) and BIC (14269.49), suggesting it is the best-fitting model. The alpha power transformed extended parameter Lindley distribution (APTEPLD) follows with higher AIC (14562.45) and BIC (14658.26) values, indicating it is less optimal than APTLD. The three-parameter Lindley distribution (TPLD) and the two-parameter Lindley distribution (TwPLD) have significantly higher AIC and BIC values, making them the least suitable models for the data. Overall, AIC and BIC help in selecting the most appropriate model by considering both fit and complexity, with APTLD emerging as the best choice for the data. More information can be found in S1 Appendix. R code.

The variables considered in Table 4 were: $y$—cognitive achievement in mathematics skill scores—as the dependent variable, along with the following explanatory variables: $x_1$(place of residence: 1=urban, 0=rural),_$x_2$ (mode of transportation: 1=public transportation, 0=walk), $x_3$ (household size), $x_4$ (electricity access: 1=yes, 0=no), $x_5$ (region of students: 1=Amhara, 2=Oromiya, 3=SNNPR, 4=Addis Ababa, 0=Tigray), $x_6$ (mother's educational level: 1=primary, 2=secondary, 3=diploma or higher, 0=no formal education), and $x_7$ (father's educational level: 1=primary, 2=secondary, 3=diploma or higher, 0=no formal education).

Table 4 presents the estimates of regression models for cognitive achievement in mathematics skill scores, analyzed using different probability distributions: APTLD, APTEPLD, TPLD, and TwPLD. The model selection criteria, AIC and BIC, for each model are as follows: APTLD (AIC=14,188.23, BIC=14,291.85), APTEPLD (AIC=14,562.45, BIC=14,658.26), TPLD (AIC=31,892.3, BIC=31,794.1), and TwPLD (AIC=63,845.6, BIC=63,752.9). These values provide a basis for comparing the models in terms of their relative goodness of fit to the data. The results indicate that the APTLD regression model has the lowest AIC and BIC values among the four models, suggesting that it provides the best fit for the data. The

**Table 3. The MLEs of the models for cognitive mathematics skill score.**

| Model | $\hat{\alpha}$ | $\hat{\theta}$ | $\hat{\delta}$ | $\hat{\beta}$ | AIC | BIC |
|---|---|---|---|---|---|---|
| APTLD | 5.74 | 2.27 | 11.14 | 7.48 | 14182.23 | 14,269.49 |
| APTEPLD | 0.49 | 0.58 | 1.02 | 0.32 | 14562.45 | 14,658.26 |
| TPLD | – | 4.43 | 2.29 | 4.65 | 16,592.3.3 | 16,694.1 |
| TwPLD | – | 2.92 | – | 2.97 | 16,933.3 | 17,009.65 |

**Table 4. Results of the fitted regression models.**

| Parameters | APTLD | | APTEPLD | | TPLD | | TwPLD | |
|---|---|---|---|---|---|---|---|---|
| | Estimate | P-value | Estimate | p-value | Estimate | P-value | Estimate | p-value |
| $\beta_0$ | 6.09 | 0.00 | 0.04 | 0.25 | −2.29 | 0.13 | −12.36 | 0.42 |
| $\beta_1$ | 9.52 | 0.02 | 0.33 | 0.00 | −6.77 | 0.00 | 2.71 | 0.07 |
| $\beta_2$ | 6.57 | 0.04 | −0.16 | 0.12 | −3.11 | 0.06 | −2.83 | 0.49 |
| $\beta_3$ | -0.15 | 0.01 | −0.14 | 0.03 | 2.71 | 0.01 | −4.73 | 0.00 |
| $\beta_4$ | 2.94 | 0.02 | −0.47 | 0.02 | −5.75 | 0.09 | 2.85 | 0.01 |
| $\beta_5$ | 7.39 | 0.06 | 0.06 | 0.32 | 2.64 | 0.08 | 0.82 | 0.71 |
| $\beta_6$ | 4.61 | 0.03 | 0.05 | 0.45 | 10.31 | 0.01 | 2.45 | 0.04 |
| $\beta_7$ | 1.67 | 0.00 | 0.02 | 0.92 | 1.69 | 0.02 | 2.07 | 0.47 |
| $\beta_8$ | 5.74 | 0.00 | 0.27 | 0.03 | −6.91 | 0.00 | −7.42 | 0.00 |
| $\beta_9$ | 1.71 | 0.07 | −0.12 | 0.08 | 1.91 | 0.04 | −2.32 | 0.04 |
| $\beta_{10}$ | 3.81 | 0.01 | −0.15 | 0.01 | 5.29 | 0.00 | 4.80 | 0.00 |
| $\beta_{11}$ | 1.12 | 0.02 | −0.23 | 0.00 | −2.76 | 0.00 | −2.91 | 0.02 |
| $\beta_{12}$ | 3.09 | 0.20 | 0.004 | 0.92 | 2.14 | 0.03 | 5.31 | 0.03 |
| $\beta_{13}$ | 5.21 | 0.01 | −0.011 | 0.87 | −3.63 | 0.04 | −2.51 | 0.04 |
| $\beta_{14}$ | 2.29 | 0.00 | −0.015 | 0.65 | −1.68 | 0.25 | −1.06 | 0.90 |
| AIC | 14,188.23 | | 14,562.45 | | 31,892.3 | | 63,845.6 | |
| BIC | 14,291.85 | | 14,658.26 | | 31,794.1 | | 63,752.9 | |

lower values of AIC and BIC for APTLD confirm its superiority over the APTEPLD, TPLD, and TwPLD models. This implies that the APTLD distribution better captures the underlying structure of the mathematics skill score data and is the most suitable model for analyzing cognitive achievement in this context.

The coefficient estimates for explanatory variables in the regression models are presented in Table 4 below. Therefore, it is clear that APTLD regression model outperforms among others for these data. According to results of a new APTLD regression model, $\beta_0$ $\beta_1$, $\beta_2$, $\beta_3$, $\beta_4$, $\beta_5$, $\beta_6$ and $\beta_7$ are statistically significant at 5% level. The estimated coefficients for various explanatory, include place of residence, transportation used by students, household size, access to electricity, and parents' educational levels (primary, secondary, and diploma and above) for both mother and father educational level with no formal education used as reference category. The region of students (Amhara, Oromiya, SNNPR, and Addis Ababa) with Tigray as the reference category. The coefficient for the place of residence (9.52) indicates that living in an urban area is associated with higher scores in cognitive achievement in mathematics skills score compared to rural areas. This suggests that urban environments may provide better educational resources and opportunities, which positively impact students' academic performance.

The estimated coefficient for transportation, where students use public buses, is 6.57. This indicates that students who use transportation (public or school transport) tend to have cognitive achievement in mathematics skills than those who

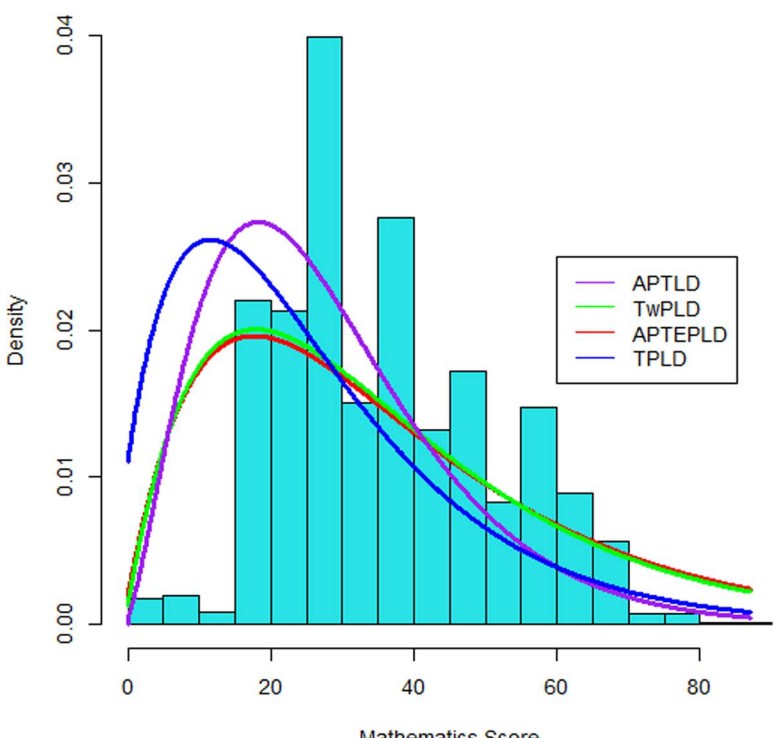

**probability density function**

**Fig 1. Fitted densities of the distributions for the mathematics skill score data.** The Fig 1 illustrates the density plot and facilitates a comparison between the fitted densities of the models and the empirical histogram of the observed data. Notably, the fitted density of the new regression model based on the APTLD distribution (represented by the purple line) demonstrates a closer alignment with the empirical histogram than the other distributions.

walk. The implication here is that longer travel times or reduced time available for studying due to transportation issues may negatively affect students' cognitive achievements.

Additionally, a larger household size is associated with lower cognitive achievement, as indicated by the significant negative coefficient (−0.15). A significant negative coefficient indicates that larger household sizes are strongly associated with lower cognitive achievement in mathematics skills score. This relationship may reflect greater resource constraints and less individual attention available to each student in larger households, thereby negatively impacting their cognitive achievement in mathematics skills score. It indicates as the household size increases by a unit, cognitive achievement in mathematics skills score decreased by 0.15%.

The coefficient for access to electricity is estimated at 2.94. A positive coefficient in this context indicates that having access to electricity is associated with higher cognitive achievement in mathematics skills score. Access to electricity likely facilitates a better study environment and access to educational resources such as the internet and proper lighting, thus positively impacting cognitive achievement in mathematics skills score.

Regarding mother's education, the estimated coefficients for primary, secondary, and diploma education levels are 7.39, 4.61, and 1.67, respectively. These positive coefficients indicate that having a mother with primary, secondary, or diploma education is associated with higher cognitive achievement in mathematics skills score compared to having a mother with no formal education. This highlights the important role of maternal education in enhancing cognitive achievement in mathematics skills score.

For father's education, the estimated coefficients for primary, secondary, and diploma education levels are 5.74, 1.71, and 3.81, respectively. These positive coefficients suggest that having a father with primary, secondary, or diploma education is associated with an increase in cognitive achievement in mathematics skills score by 5.74%, 1.71%, and 3.81%, respectively, compared to the reference category of no formal education.

The coefficients for the regions, with Tigray as the reference category, indicate varying impacts on cognitive achievement in mathematics skill score. Specifically, being from the Amhara region (1.12) is associated with a slight increase, while the Oromia region (3.09) shows a more substantial positive effect. The SNNPR region (5.21) has the highest positive coefficient, suggesting the most significant positive impact. Meanwhile, the Addis Ababa region (2.29) indicates a moderate increase. These coefficients highlight regional differences, with SNNPR showing the most substantial positive effect on students' cognitive achievement in mathematics compared to Tigray region. (See S1 Appendix. R code for more information on how it was obtained).

## Discussion

This study aims at identifying the best probability distributions for students' cognitive achievements in mathematics skill score, including APTLD, and analyzing the Young Lives data sets in Ethiopia. Accordingly, the average cognitive achievement in mathematics skill scores was 37%. The results of this study are comparable to those of [10–13], which were lower than the current study. Students from urban areas attending private schools have better cognitive achievement in the mathematics skill score indicators, and this finding was similar to the study done by [13]. The cognitive achievement in the mathematics skill score is better predicted by the parent's educational background. It reveals that mothers having primary, secondary, diploma, and above educational levels have better cognitive achievement skill scores compared to mothers with no formal education as well as fathers with educational backgrounds. These results are supported by [14,16], and [18] authors who basically focused on science skill scores.

Regression models were used in the current study to apply the APTLD and other relative distributions; the findings show that, when compared to the other stated distributions, the APTLD regression models fit the data the best. These findings were similar to those of authors [17–22], who used regression models on mathematics skill scores to examine various distribution types.

### Limitation of the study

The study primarily focuses on Young Lives datasets, limiting its applicability to broader student populations and different educational contexts. Additionally, factors such as school quality, teacher experience, resources, parental employment, income levels, and socioeconomic status are not explicitly considered. Future research could enhance the APTLD regression model using Bayesian methods to evaluate its robustness across diverse datasets and its effectiveness in modeling external influences on cognitive achievement.

### Conclusions

In this study, the application of the alpha power transformation Lindley distribution (APTLD) on cognitive achievements in mathematics skill scores using regression model framework was developed. Various statistical probability distributions have been examined. The distribution has demonstrated its importance in real-life situations, such as in probability distribution regression models. The APTLD has provided significant flexibility for new applications in education and has been thoroughly studied. The APTLD regression model is a specific type of regression model that assumes the response variable follows the APTLD distribution. Based on the results of these applications, the study found that students' average mathematics skill score was 37.01%, with a standard deviation of 14.9, reflecting performance variation. Parental education differed significantly, with 48.5% of mothers and 34% of fathers lacking formal schooling. Additionally, 59% of students lived in rural areas, while 41% resided in urban settings. The average household size

was 5.77 members, showing variability in family structures. From the results, the findings show that the mean cognitive achievement in mathematics skill scores (37.01) is greater than the median (33.33), indicating that the data are positively skewed or right-skewed.

According to the results of APTLD regression model, place of residence, mode of transportation, household size, electricity access, region of students, and parent educational levels are statistically significant at 5% level. We can conclude that the APTLD regression model demonstrates the best fit for the data, as indicated by its lowest AIC and BIC values compared to the APTEPLD, TPLD, and TwPLD models. This confirms its superiority in capturing the underlying structure of mathematics skill scores, making it the most suitable model for analyzing cognitive achievement. the APTLD distribution is characterized by its heavy tails, allowing for extreme values in the data. Therefore, this new model can be considered a significant contribution to the field of statistics and probability methods. Future research could enhance the APTLD regression model using Bayesian methods to evaluate its robustness across diverse datasets and its effectiveness in modeling external influences on cognitive achievement.

## Supporting information

**S1 Appendix. R code.**
(DOCX)

## Acknowledgments

We would like to thank the Young Lives program, for providing the dataset used in this study. The authors would like to extend their sincere gratitude to Kotebe University of Education for their continuous support of this study.

## Author contributions

**Conceptualization:** Shibiru Jabessa Dugasa, Butte Gotu Arero.

**Data curation:** Shibiru Jabessa Dugasa.

**Formal analysis:** Shibiru Jabessa Dugasa, Butte Gotu Arero.

**Investigation:** Shibiru Jabessa Dugasa.

**Methodology:** Shibiru Jabessa Dugasa, Butte Gotu Arero.

**Software:** Shibiru Jabessa Dugasa.

**Supervision:** Butte Gotu Arero.

**Validation:** Butte Gotu Arero.

**Visualization:** Shibiru Jabessa Dugasa, Butte Gotu Arero.

**Writing – original draft:** Shibiru Jabessa Dugasa, Butte Gotu Arero.

**Writing – review & editing:** Shibiru Jabessa Dugasa, Butte Gotu Arero.

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
