## [Decision Letter · Decision Letter 0]

Thank you for submitting your manuscript to PLOS ONE. After careful consideration, we feel that it has merit but does not fully meet PLOS ONE’s publication criteria as it currently stands. Therefore, we invite you to submit a revised version of the manuscript that addresses the points raised during the review process.

We look forward to receiving your revised manuscript.

Kind regards,

Daniel Biftu Bekalo, PhD

Academic Editor

PLOS ONE

Journal Requirements:

Reviewers' comments:

Reviewer's Responses to Questions

**Comments to the Author**

1. Is the manuscript technically sound, and do the data support the conclusions?

Reviewer #1: Yes

Reviewer #2: Partly

2. Has the statistical analysis been performed appropriately and rigorously?

Reviewer #1: Yes

Reviewer #2: Yes

3. Have the authors made all data underlying the findings in their manuscript fully available?

Reviewer #1: Yes

Reviewer #2: Yes

4. Is the manuscript presented in an intelligible fashion and written in standard English?

Reviewer #1: Yes

Reviewer #2: No

Reviewer #1: In this paper, the authors have considered the the distribution an extended Lindley distribution via obtained from alpha power alternative. The manuscript under review is interesting and documented, including both theoretical and practical results. The bibliography shows the authors' adequate documentation. However, I consider that the paper will benefit if the authors address within the manuscript the following aspects:

General remarks regarding the paper. The authors must assume more clearly in the paper their original contribution by specifying this fact and by highlighting the fact that starting from a certain point there are presented the original and novel aspects of their research. The authors must state more clearly their original methods, their original results and conclusions, the novelty of their study.

Remark regarding the "Abstract" of the paper. I consider that the authors should structure the "Abstract" as to cover the most important points of interest: the authors should have positioned the manuscript's topic in a broad context therefore covering appropriately the topic's background; the authors should have presented succinctly the methods they have employed in order to attain the purpose of their study; the authors should have summarized the most important outcomes of their study and the conclusions that one could draw. I consider that the abstract of the manuscript under review will be improved if the authors state and explain concisely at the end of the abstract the clear contribution that their study has brought to the current state of knowledge. In the abstract, the authors must state more clearly for the reader what is the meaning, the purpose, the usefulness of the research developed within the paper, what are their methods, their original results and conclusions as well as the novelty of their study.

Remarks regarding the results of the paper. The authors should in detail present the findings and their main implications, also highlighting current limitations of their study, and briefly mention some precise directions that they intend to follow in their future research work. I consider that the paper will benefit if the authors make a step further, beyond their analysis and provide an insight at the end of this section regarding what they consider to be, based on the obtained results, the most important, appropriate and concrete actions that the decisional factors and all the involved parties should take in order to benefit from the results of the research conducted within the manuscript.

Remark regarding the "Conclusions" section. I consider that it will benefit the manuscript and at the same time it will highlight even more the authors' contribution if they provide an insight stating clearer what is the purpose and usefulness of their study. Secondly, I consider that the authors should provide more details regarding the domains in which their modeling strategy can be applied, because it is not suitable to put the reader in the situation of interpreting, analyzing, continuing or refining the study from the manuscript under review.

Introduction Section should be improved via newly defined Lindley type distributions such as:

* "Modified-Lindley distribution and its applications to the real data." Communications Faculty of Sciences University of Ankara Series A1 Mathematics and Statistics 71, no. 1 (2022): 252-272.

*"An alternative distribution to Lindley and Power Lindley distributions with characterizations, different estimation methods and data applications." Mathematica Slovaca 70, no. 4 (2020): 953-978.

*A new alternative unit-Lindley distribution with increasing failure rate.Scientia Iranica, 2025. DOI: 10.24200/SCI.2022.58409.5712

*Some theoretical and computational aspects of the odd Lindley Fréchet distribution.İstatistikçiler Dergisi: İstatistik ve Aktüerya 10, no. 2 (2017): 129-140.

*The xgamma family: Censored regression modelling and applications. Revstat-Statistical Journal 18, no. 5 (2020): 593-612.

*"The odd Lindley Nadarajah-Haghighi distribution." J. Math. Comput. Sci. 7, no. 5 (2017): 864-882.

*"The quasi XGamma-Poisson distribution: properties and application." Istatistik Journal of The Turkish Statistical Association 11, no. 3 (2018): 65-76.

*"The Type II quasi lambert g family of probability distributions." Pakistan Journal of Statistics and Operation Research (2022): 963-983.

*Odd Lindley-Lomax model: Statistical properties and applications." Pakistan Journal of Statistics and Operation Research (2019): 419-430.

* "The odd Lindley Burr XII model: Bayesian analysis, classical inference and characterizations." Journal of data science (2018).

Reviewer #2: Referee Report on

“Modelling Students’ Cognitive Achievements in Mathematics Using the Alpha Power Lindley Probability Distribution”

This paper is good prepared and organized. It proposes regression modeling using the Alpha Power Transformation Lindley (APTLD) probability distribution for the application of cognitive achievement in mathematics skill scores. The study also compares this distribution with other competitive probability distributions. Young Lives data was used for the present study, and different explanatory variables were included. The results show that the mean cognitive achievement in mathematics skill scores (37.01) is greater than the median (33.33), indicating that the data are positively skewed or right skewed. The comparison of the fitted models suggests APTLD as the best model.

I recommend accepting this paper after major revisions:

1) Moderate English changes required.

2) The author should write the motivation of this study clearly.

3) The authors must rewrite the introduction and add the section titles and summary about them.

4) The authors must proof the APTLD follows an exponential family before start doing the GLM.

5) The authors must include the direct link for the dataset used in this study.

6) In probability, we always use capital letters to denote the random variables. Also, the estimated parameter with add a cap up of it, for example β ^. So, the authors must make this change

7) η_i is equation (2) should be changed to X_i β.

8) I recommend adding the R codes to the appendix.

**Do you want your identity to be public for this peer review?** For information about this choice, including consent withdrawal, please see our Privacy Policy

Reviewer #1: No

Reviewer #2: No

---

## [Decision Letter · Decision Letter 1]

Dear Dr.  Dugasa,

Thank you for submitting your manuscript to PLOS ONE. After careful consideration, we feel that it has merit but does not fully meet PLOS ONE’s publication criteria as it currently stands. Therefore, we invite you to submit a revised version of the manuscript that addresses the points raised during the review process.

We look forward to receiving your revised manuscript.

Kind regards,

Daniel Biftu Bekalo, PhD

Academic Editor

PLOS ONE

Additional Editor Comments:

Dear Authors, Try to incorporate the reviewer's comments. 

Reviewers' comments:

**Comments to the Author**

Reviewer #1: All comments have been addressed

Reviewer #2: (No Response)

2. Is the manuscript technically sound, and do the data support the conclusions?

Reviewer #1: Yes

Reviewer #2: Yes

3. Has the statistical analysis been performed appropriately and rigorously?

Reviewer #1: Yes

Reviewer #2: No

4. Have the authors made all data underlying the findings in their manuscript fully available?

Reviewer #1: Yes

Reviewer #2: Yes

5. Is the manuscript presented in an intelligible fashion and written in standard English?

Reviewer #1: Yes

Reviewer #2: Yes

Reviewer #1: The authors have applied comments. Hence, the paper can be recommended for the possible publication in the journal.

Reviewer #2: Referee Report on

“Modelling Students’ Cognitive Achievements in Mathematics Using the Alpha Power Lindley Probability Distribution”

I recommend accepting this paper after major revisions:

1. The authors must rewrite the introduction and add the section titles and summary about them by mentioning that section by section with section number.

2. The authors must prove the APTLD follows an exponential family before starting doing the GLM.

3. I recommend adding the R codes to the appendix (I can’t see the appendix in the modified version).

**Do you want your identity to be public for this peer review?** For information about this choice, including consent withdrawal, please see our Privacy Policy

Reviewer #1: No

Reviewer #2: No

---

## [Author Response · Author response to Decision Letter 2]

26 Jun 2025

I have attached the reviewers respond

---

## [Decision Letter · Decision Letter 2]

Modelling students’ cognitive achievement skills using the alpha power transformed Lindley probability distribution

PONE-D-25-16740R2

Dear Dr. Dugasa,

We’re pleased to inform you that your manuscript has been judged scientifically suitable for publication and will be formally accepted for publication once it meets all outstanding technical requirements.

Kind regards,

Srinivasa Rao Gadde, Ph.D.

Academic Editor

PLOS ONE

Additional Editor Comments (optional):

The manuscript may be accepted. Before submitting the final version, the author has to correct the language. Some sentences are incorrect and need to be checked once.

Reviewers' comments:

Reviewer's Responses to Questions

**Comments to the Author**

Reviewer #1: (No Response)

Reviewer #2: All comments have been addressed

2. Is the manuscript technically sound, and do the data support the conclusions?

Reviewer #1: (No Response)

Reviewer #2: Yes

3. Has the statistical analysis been performed appropriately and rigorously?

Reviewer #1: (No Response)

Reviewer #2: Yes

4. Have the authors made all data underlying the findings in their manuscript fully available?

Reviewer #1: (No Response)

Reviewer #2: Yes

5. Is the manuscript presented in an intelligible fashion and written in standard English?

Reviewer #1: (No Response)

Reviewer #2: (No Response)

Reviewer #1: The authors have applied comments about suggestions. Hence, I recommend the paper for the possible publication in the journal. Surely, the last decision belongs to editor.

Reviewer #2: Referee Report on

“Modelling Students’ Cognitive Achievements in Mathematics Using the Alpha Power Lindley Probability Distribution”

I have checked the revised paper again, now the manuscript has been sufficiently improved to warrant publication in PLOS ONE.

**Do you want your identity to be public for this peer review?** For information about this choice, including consent withdrawal, please see our Privacy Policy

Reviewer #1: No

Reviewer #2: No

---

## [Editor Report · Acceptance letter]

PONE-D-25-16740R2

PLOS ONE

Dear Dr. Dugasa,

I'm pleased to inform you that your manuscript has been deemed suitable for publication in PLOS ONE. Congratulations! Your manuscript is now being handed over to our production team.

Kind regards,

on behalf of

Professor Srinivasa Rao Gadde

Academic Editor

PLOS ONE